# Grains, trade and war in the multimodal transmission of Rice yellow mottle virus: An historical and phylogeographical retrospective

Innocent Ndikumana[1�], Geoffrey Onaga[2�], Agnès Pinel-Galzi[3], Pauline Rocu[4],
Judith Hubert[5,6], Hassan Karakacha Wéré[7], Antony Adego[7], Mariam Nyongesa Wéré[7],
Nils Poulicard[3], Maxime Hebrard[8], Simon Dellicour[9,10,11], Philippe Lemey[10],
Erik Gilbert[12], Marie-José Dugué[13], François Chevenet[14], Paul Bastide[15],
Stéphane Guindon[4], Denis Fargette[3�*], Eugénie Hébrard[3�*]

1 Rwanda Agriculture Board, Kigali, Rwanda, 2 Africa Rice Center, Bouaké, Cote d'Ivoire, 3 PHIM Plant Health Institute, Univ Montpellier, IRD, CIRAD, INRAE, Institut Agro, Montpellier, France, 4 Department of Computer Science, Laboratoire d'Informatique, de Robotique et de Microélectronique de Montpellier, CNRS and Université de Montpellier, Montpellier, France, 5 Department of Crop Science and Horticulture, Morogoro, Tanzania, 6 World Vegetable Center, Eastern and Southern Africa, Arusha, Tanzania, 7 Masinde Muliro University of Science and Technology, Kakamega, Kenya, 8 Genome Institute of Singapore, Agency for Science, Technology and Research, Singapore, Singapore, 9 Spatial Epidemiology Lab, Université Libre de Bruxelles, Brussels, Belgium, 10 Department of Microbiology, Immunology and Transplantation, Rega Institute, KU Leuven, Leuven, Belgium, 11 Interuniversity Institute of Bioinformatics in Brussels, Université Libre de Bruxelles, Vrije Universiteit Brussel, Brussels, Belgium, 12 Department of History, Arkansas State University State University, Jonesboro, Arkansas, United States of America, 13 Agronomy and Farming Systems, Saint Mathieu de Tréviers, France, 14 MIVEGEC, IRD, CNRS, Université de Montpellier, Montpellier, France, 15 IMAG, Université de Montpellier, CNRS, Montpellier, France

☉ These authors contributed equally to this work.
* denis.fargette@ird.fr (DF); eugenie.hebrard@ird.fr (EH)

## Abstract

Rice yellow mottle virus (RYMV) is a major pathogen of rice in Africa. RYMV has a narrow host range limited to rice and a few related poaceae species. We explore the links between the spread of RYMV in East Africa and rice history since the second half of the 19th century. The phylogeography of RYMV in East Africa was reconstructed from coat protein gene sequences (ORF4) of 335 isolates sampled over two million square kilometers between 1966 and 2020. Dispersal patterns obtained from ORF2a and ORF2b, and full-length sequences converged to the same scenario. The following imprints of rice cultivation on RYMV epidemiology were unveiled. RYMV emerged in the middle of the 19th century in the Eastern Arc Mountains where slash-and-burn rice cultivation was practiced. Several spillovers from wild hosts to cultivated rice occurred. RYMV was then rapidly introduced into the nearby large rice growing Kilombero valley and Morogoro region. Harvested seeds are contaminated by debris of virus infected plants that subsist after threshing and winnowing. Long-distance dispersal of RYMV is consistent (i) with rice introduction along the caravan routes from the Indian Ocean Coast to Lake Victoria in the second half of the

**Data availability statement:** Nucleotidic sequences and associated metadata used in this study are available in the NCBI database, accession numbers are in S1-S4 Tables. All other relevant data are within the manuscript and its Supporting Information files.

**Funding:** This work was partly supported by the French National Research Agency as an "Investissements d'avenir" program (ANR-10-LABX-001-01 Labex Agro) coordinated by Agropolis Foundation (project no. 1504-004 E-SPACE to IN, EPG, NP, DF, EH) and by a bilateral project between Kenya and France (PHC PAMOJA no 36128PK to HKW, AA, MNW, EH) cofunded by National Commission for Science, Technology and Innovation (NACOSTI) and Ministère de l'Europe et des Affaires Etrangères (MEAE). PR's internship at the University of Montpellier was founded by the I-SITE MUSE through the Key Initiative "Data and Life Sciences". SD acknowledges support from the Fonds National de la Recherche Scientifique (F.R.S.-FNRS, Belgium; grant n°F.4515.22), from the Research Foundation - Flanders (Fonds voor Wetenschappelijk Onderzoek - Vlaanderen, FWO, Belgium; grant n°G098321N), from the European Union Horizon 2020 projects MOOD (grant agreement n°874850) and LEAPS (grant agreement n°101094685). GO acknowledges support from the Plant Health Initiative (PHI) funded by the CGIAR Trust Fund. The funders had no role in the study design, data collection and interpretation, or the decision to submit the work for publication.

**Competing interests:** The authors have declared that no competing interests exist.

19th century, (ii) seed movement from East Africa to West Africa at the end of the 19th century, from Lake Victoria to the north of Ethiopia in the second half of the 20th century and to Madagascar at the end of the 20th century, (iii) and, unexpectedly, with rice transport at the end of the First World War as a troop staple food from the Kilombero valley towards the South of Lake Malawi. Overall, RYMV dispersal was associated to a broad range of human activities, some unsuspected. Consequently, RYMV has a wide dispersal capacity. Its dispersal metrics estimated from phylogeographic reconstructions are similar to those of highly mobile zoonotic viruses.

---

## Author summary

Rice yellow mottle virus (RYMV) poses a major threat to rice production in Africa. We explored through a multidisciplinary approach the links between the history of rice in East Africa since the second half of the 19th century and the spread of RYMV. The results illuminate the causes of RYMV diffusion. We show the role of long-distance caravan trade, the impact of the First World War and the consequences of seed exchange in the dispersal of RYMV. The paradoxical role of seeds in the spread of RYMV - which is vector transmitted and not seed transmitted – is explained in the light of rice biology and agronomy. Overall, this study reveals the wide range of transmission ways, some unexpected, in the dispersal of plant viruses. It also highlights the role of human transmission of pathogens and sheds light on the risk of transmission of RYMV and of other plant viruses from Africa to other continents.

## Introduction

Contact transmission is a major transmission pathway for plant viruses, yet it has largely been neglected [1]. Contact-transmitted viruses have particles that are stable outside the infected cells, remain infectious on contaminated surfaces, and reach high concentrations (titer) in infected plants. Owing to their virion stability and high titer, Sobemoviruses (*Solemoviridae* family), Tobamoviruses (*Virgaviridae*), Potexviruses (*Alphaflexiviridae*) and Tombusviruses (*Tombusviridae*) are listed as candidates for contact and abiotic transmission [1]. The sobemovirus genus differs from the three other genera by having a narrow host range and by being transmitted by insect vectors. Beetles are the most frequently listed in sobemovirus transmission. Beetles are also involved in the transmission of several other genera [2]. This study aimed at evaluating the relative importance of abiotic and vector-mediated transmission in plant virus species with a dual mode of transmission (i.e., vector and abiotic). We hypothesized that the contribution of abiotic transmission to plant virus spread has been substantially underestimated [1,2]. Rice yellow mottle virus (RYMV), which affects rice crops in Africa, is an appropriate model to test this hypothesis.

Rice yellow mottle virus (RYMV) is a single-stranded RNA virus of the *Sobemovirus* genus [3]. RYMV is a major pathogen in all sub-Saharan African countries that grow rice [4]. Its host range is limited to the two cultivated rice species *Oryza sativa* and *O. glaberrima*, the wild rice species and a few related *Eragrostideae* species [5]. RYMV does not infect the seed embryo [6,7,8]. RYMV is transmitted by chewing insects, mainly beetles of the *Chrysomelidae* family in a non-persistent mode, and by leaf and root contact between infected and healthy plants [5]. RYMV reached high virus concentration within two to three weeks. It remains infectious in rice stubbles for months [5,7], and after passage through the intestinal tract of mammals [9]. The spread of the disease was first attributed to vector transmission, although no clear link has been found between the size of the beetle populations and the incidence of the disease. Later, the role of mechanical transmission through leaf-to-leaf and root-to-root contact was established experimentally [10]. Contamination in the soil also occurred by contact of the roots of young seedlings with infected rice stubble present in the soil. This is referred to as soil transmission or soil contamination [1,11,12]. Soil contamination has several origins, infected rice residues buried in the soil after harvesting or after mammal consumption, and also by planting rice seeds contaminated by infected plant residues (see below). Due to the stability of RYMV [7], soil transmission persists months after the infected rice residues have been buried. RYMV is disseminated by transplanting young infected seedlings from nurseries to fields sometimes located a few kilometers away [13]. Altogether, contact and soil transmission accounts for on-site survival of the virus and its local spread.

Yet, the long-distance transmission of RYMV has not been elucidated. Without experimental evidence, it has been tentatively ascribed to movements of beetles during tornadoes, transmission by more proficient flyer insects such as grasshoppers [5], or transmission by seeds in specific circumstances [14,15]. The last of the three hypotheses, which posited transmission with seeds, relied on the following observations. RYMV is not present in the rice embryo but is detected in the glumes, and in the outer envelopes of the grain [8]. Rice threshing and winnowing does not eliminate all the outer husks and all the leaf and stem residues [16]. Accordingly, infected plant debris remain in bags of rice seeds. We referred to them as contaminated seeds to make the distinction with infected seeds [4,17]. Rice seeds are transported between villages and across regions for food consumption and for agronomical purposes [18]. Rice planting with contaminated seeds as well as consumption of contaminated rice seeds by mammals (including humans) result in burying infected rice residues in the soil. Rice planted in a contaminated soil is infected through mechanical contact of the roots of the young seedlings with the infected residues. Based on these observations, it was hypothesized that RYMV was transported with bags of contaminated rice seeds to new areas, potentially far away from its sources [4,17]. Accordingly, seed transport would be the primary mode, possibly the exclusive mode, of long distance dissemination of RYMV.

Nevertheless, long-distance transmission of plant viruses, be it by vectors or with seeds, has been difficult to establish [19,20], and this is true for RYMV as well. None of the hypotheses put forward for long-distance transmission of RYMV has been experimentally validated. Phylogeography offers new perspectives through the reconstruction of the spatiotemporal pattern of spread [21,22]. It unveils the underlying processes that shape the dynamics of spread, including rare and past events, and lead to hypotheses on the modes of transmission and the impact of host biology and history. RYMV is a measurably evolving population [23], and the timescale of the phylogeographic reconstructions is not affected by the time dependent rate phenomenon [24]. The relaxed random walk (RRW) model in continuous phylogeography allows for variation in dispersal rates across branches of the phylogeny [22]. This model provides flexibility to accommodate the different means of transmission when reconstructing the phylogenetic dispersal history of the virus.

In this study, the phylogeography of RYMV in East Africa was reconstructed from a comprehensive heterochronous sequence dataset. Given that RYMV has a narrow host range limited to rice and a few related species, it was expected that its dissemination has been shaped by rice history. Subsequently, we paid special attention to the history of rice in East Africa since the beginning of the 19th century. RYMV originates in East Africa but unlike in Madagascar and in West Africa, rice cultivation until the last decades remained mainly confined to lake shores, river banks and wetlands [25, 26, 27]. Rice had been cultivated along the Indian Ocean Coast for centuries but rice cultivation was restricted to a few scattered coastal sites [26]. In the second half of the 19th century, large plantations of rice were reported in the

Morogoro region [28] and documented in the Kilombero valley, upstream of the mouth of the coastal river Rufiji, 300 km from the Indian Ocean [29]. By the end of the 19th century, rice supplies from the Rufiji flood plains had emerged as a sufficiently important source for that area to acquire the nickname "Little Calcutta" [30,31,32]. In the middle of the 19th century, rice was introduced around the great lakes of East Africa (Lake Victoria, Lake Tanganyika and Lake Malawi) [33]. Rice cultivation became generalized at the end of the 20th century. Assessing the virus movements between these far apart rice-producing regions helps to identify the main modes of dissemination.

Dispersal statistics have been increasingly used to characterize and quantify the spread of viruses [34,35,36]. We assessed the dispersal capacity of RYMV and compare it to a range of zoonotic viruses [35]. Altogether, this study suggests that the transport of contaminated rice seed was decisive in the long-distance dissemination of RYMV, not only within East Africa, but also in the introduction of the virus from East Africa to West Africa and to Madagascar. It points the most probable routes and modes of dissemination and unveils the links with rice history. It further shows the role of a broad spectrum of human activities in RYMV dispersal, some of them unsuspected. This explains why RYMV exhibits extensive dissemination capabilities, similar to those of highly mobile zoonotic viruses.

## Results

### Temporal signal

BETS is a Bayesian evaluation method of the temporal signal which involves comparing the fit to the data of two models: a model in which the data are accompanied by the actual (heterochronous) sampling times, and a model in which the samples are constrained to be contemporaneous (isochronous) [37]. BETS analyses indicate that the increase in sample size of heterochronous sequences lead to a stronger temporal signal in West Africa (from 261 to 335 sequences) and in East Africa (from 240 to 335 sequences) (Table 1). The temporal signal was more pronounced in the set of West African sequences (reflected by a higher Bayes factor). The temporal signal of the two datasets was also tested using cluster randomization tests of the sampling dates. There was no overlap of the credibility intervals of the substitution rate calculated with sampled dates and those with randomized dates. However, the differences obtained were more pronounced with the WA335 dataset. The square of the correlation coefficient of the regression sampling date vs. distance to the root obtained with TreeTime was 0.07 with the WA335 dataset vs. 0.01 with the EA335 dataset. Altogether, the temporal signal was more pronounced in the WA335 dataset than in the EA335 dataset. The stronger temporal signal of the West African dataset probably results from a relatively uniform distribution of the sampling dates over the last thirty years whereas that of East African isolates was concentrated over the last 15 years. Short sampling periods are prone to unpurged transitory deleterious polymorphism leading to higher and biased substitution rates [38]. Consistently, a recent article reported that substitution rates of plant viruses decreased with increasing sampling time span [39]. Viral isolates in contaminated

**Table 1. Temporal signals in the datasets assessed by BETS.**

| Datasets | Log marginal likelihood* | | Bayes factor |
|---|---|---|---|
| | Isochronous | dated | |
| West-Africa | | | |
| WA261 | -10495 | -10409 | 86 |
| WA335 | -13546 | -13379 | 167 |
| | | | |
| East-Africa | | | |
| EA240 | -11294 | -11246 | 48 |
| EA335 | -14634 | -14539 | 95 |

*estimated under a strict molecular clock model.

seeds do not accumulate nucleotide substitutions over time. These periods of stasis, both frequent and prolonged during the initial stages of RYMV epidemiology in East Africa (see below), may introduce evolutionary rate variation among viral lineages. Altogether, the weak temporal signal of the EA335 dataset leads to a biased high substitution rate and correlatively to an inconsistent age of the most recent common ancestor (TMRCA) of the sampled sequences, similar to that in West Africa. It was obviously too recent as, given the nested position of the West African lineage within the East African lineages (see below), the TMRCA in East Africa should to be substantially older than that in West Africa.

As the temporal signal from the East African corpus was not sufficient alone to allow a valid reconstruction of the spatio-temporal dispersion of the RYMV in East Africa, priors derived from the WA335 dataset were applied. Accordingly, for the EA335 phylogeographic reconstructions, a prior was specified on the mean rate under the uncorrelated log-normal relaxed molecular clock (the "ucld.mean" parameter), a normal distribution with a mean of $1.24 \times 10^{-3}$ substitutions per site per year and standard deviation of $1.16 \times 10^{-4}$. An informative prior was also specified on the standard deviation of the uncorrelated log-normal relaxed clock (the "ucld.stdev" parameter), a normal distribution with a mean of $1.09 \times 10^{-3}$ substitution per site per year and standard deviation of $2.87 \times 10^{-4}$.

## Phylogeny of RYMV in East-Africa

The phylogeny of RYMV in East Africa reconstructed from 335 coat protein gene sequences is made of three major lineages named, respectively, S4, S5 and S6 lineages (Fig 1). The phylogenetic group containing the S4 and S5 lineages

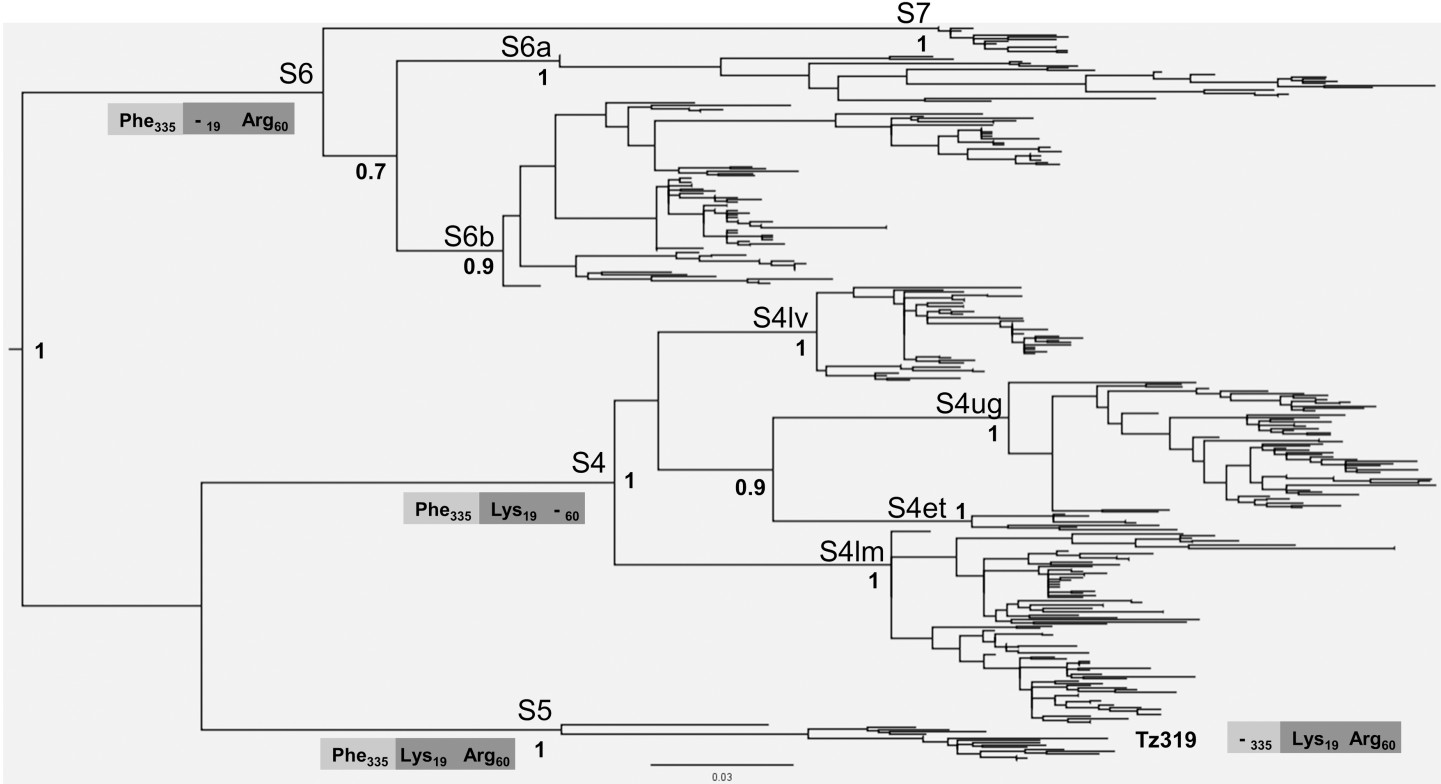

**Fig 1. Phylogeny of RYMV in East Africa.** Phylogeny of RYMV reconstructed from the coat protein gene sequences of 335 isolates through the maximum likelihood method under a HKY85 model and mid-point rooting. The names of the lineages and of the strains are given. The bootstrap support (> 0.70) is indicated at the node of the lineages and of the strains. The names of the lineages and of the strains are given. The insertion-deletion polymorphism (amino-acid and position) of the lineages and of the isolate Tz319 are framed in light grey for the ORF2a and in dark grey for the ORF4.

does not have a strong bootstrap support. Several strains are distinguished within the S4 and S6 lineages. The S4 lineage includes strains S4lm, S4lv, S4ug and S4et named according to their geographical distribution (lm, Lake Malawi; lv, Lake Victoria; ug, Uganda; et, Ethiopia). The S6 lineage includes strains S6a, S6b, and S7 but was not supported by a strong bootstrap value. The S5 lineage consists of one strain only.

Very few insertion-deletion events occurred in coding regions. The basal insertion-deletion pattern at positions 19 and 60 in the ORF4 in the S4, S5 and S6 lineages was that observed earlier [40]. An additional deletion occurred in the isolate Tz319 of the S5 strain at codon 432 in the ORF2a (Fig 1). It is the only insertion-deletion event found at a terminal phylogenetic position. All the insertion-deletion polymorphisms originated in isolates of the Eastern Arc Mountains.

### Phylogeography from 335 coat protein gene sequences

Inference using statistical phylogeographic models, including those used in the present study, may be impacted by the way samples were collected [41,42]. In particular, observations deriving from a single point in time do not convey information about a directionality in the migration patterns, making it difficult, if not impossible, to accurately reconstruct ancestral locations. Biases in sampling locations, whereby these locations follow particular geographic patterns that depart from the distribution of the whole population, may also result in biases in ancestral reconstructions. The latter will reflect sampling locations rather than the spatial distribution of ancient populations. Finally, sampled locations are the product of two processes. The first governs the random fluctuation of lineage locations during the course of evolution. The second corresponds to logistic constraints that prevent specific regions from being included in the sampled areas. Ignoring these constraints leads to overly precise estimates of the spatial diffusion parameters [42]. The present study relies however on heterochronous data, which, at least in theory, conveys information about the directionality of the migration patterns. Also, it is likely that most regions of RYMV circulation in East Africa were surveyed and incorporated in our sample. Then, the sampling area should reflect the spatial distribution of the corresponding pathogen's population. The sampling scheme here is thus clearly different from targeting a small number of specific regions, selected based on logistic considerations only. While sampling biases cannot be fully excluded, their impact on the reconstructions of RYMV spread in Africa should be limited.

The spatio-temporal dispersal of RYMV lineages in East Africa was reconstructed from the capsid protein gene sequences of 335 isolates (Fig 2A and 2B), giving rise to the following most probable historical scenario.

(1) In the middle of the 19th century (1845, 95% HPD [1788–1897]), RYMV emerged around the Great Ruaha Escarpment in the Udzungwa mountains of the Eastern Arc Mountains north of the Kilombero Valley (Fig 2C);

(2) Within a few decades, RYMV dispersed to the adjacent Kilombero Valley (Fig 2C);

(3) Between the end of the 19th century and the beginning of the 20th century, RYMV spread to the Morogoro region 180 km at the north east of the Kilombero valley. During the same period, RYMV reached the south of Lake Victoria 400 km northward, the first long range dispersal event (I) (Fig 2D);

(4) In the following decades, RYMV circulated at the east of Lake Victoria (Fig 2E);

(5) A second long range dispersal event (II) originated at the beginning of the 20th century (1920, 95% HPD [1887–1949]) in the Kilombero valley or in the Morogoro region to reach the south of Lake Malawi, 900 km southward, ca. 80 years later (Fig 2E and 2F);

(6) A third long range dispersal event (III) took place in the second half of the 20th century. It originated at the east of Lake Victoria and reached northern Ethiopia, 1400 km northward, 50 years later (Fig 2F);

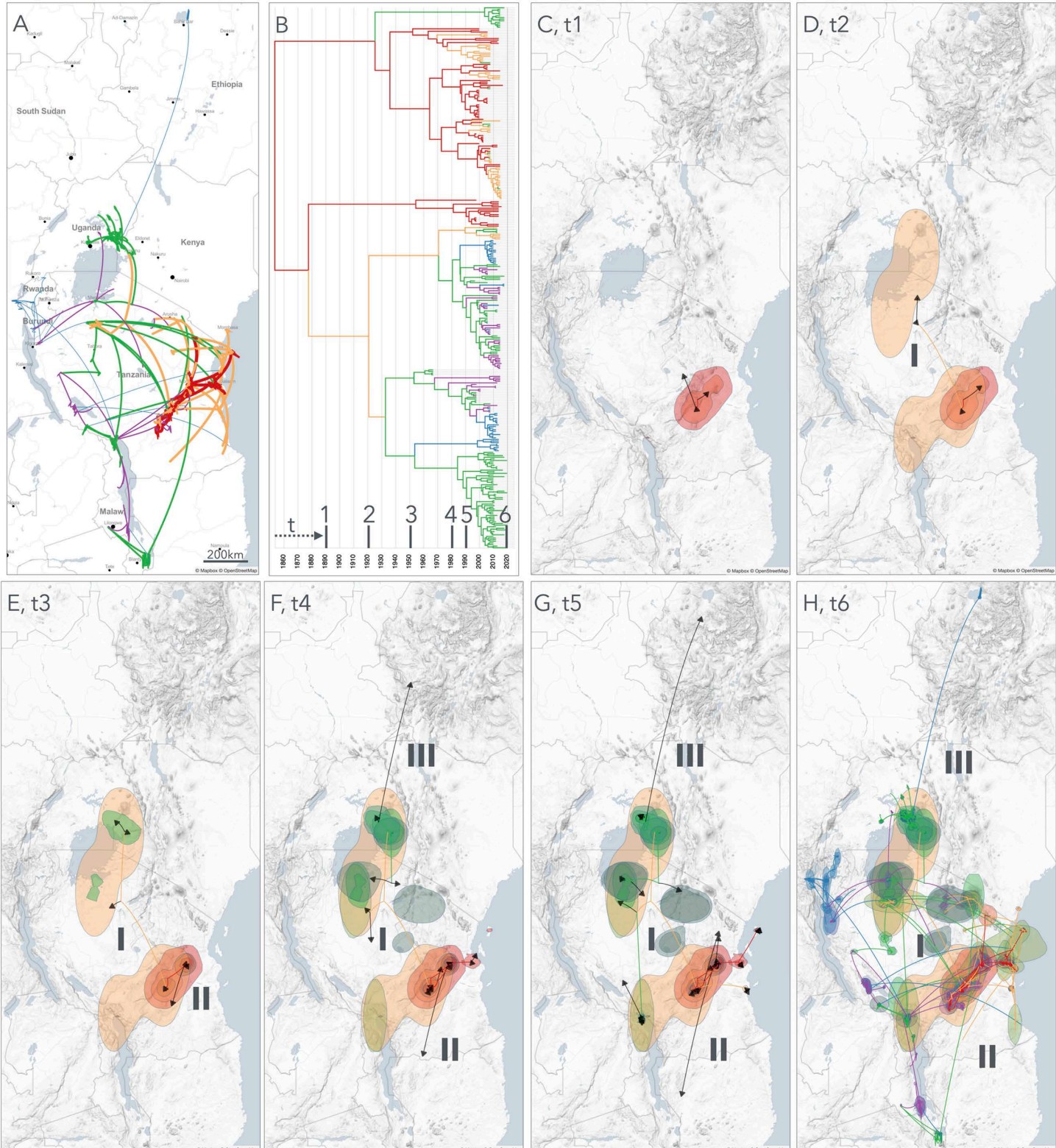

**Fig 2. Phylogeography of RYMV in East Africa.** Continuous phylogeographic reconstruction of the spatiotemporal dispersal of RYMV lineages in East Africa from 1850 to 2020 based on the coat protein gene sequences of 355 isolates collected from 1966 to 2019. **(A)** Overview of the spread. **(B)**

Time-calibrated phylogenic tree, the successive periods of dispersal are indicated (tx). **(C-H)** The maximum clade credibility (MCC) tree and the 95% highest posterior density (HPD) regions reflecting the uncertainty related to the phylogeographic inference were mapped. The phylogeographic scenario is displayed with temporal gradients linked to HPD and line colors. Ongoing movements (gradient of colors) are distinguished from completed ones (lines in black). Arrows visualize the direction of the movements. Prints use map data from Mapbox and OpenStreetMap and their data sources. To learn more, visit https://www.mapbox.com/about/maps/ and http://www.openstreetmap.org/copyright.

(7) During the last decades of the 20th century, RYMV dispersed eastward and northward from the Kilombero Valley and the Morogoro region, and southward and westward from Lake Victoria. Recently, the virus became widespread in all rice-growing regions with an increase in movements between adjacent localities (Fig 2G and 2H).

Altogether, the phylogeography of RYMV in East Africa reconstructed from the 335 coat protein gene sequences was characterized by these three long-distance dispersal events along a south-north axis from the middle of the 19th century to the second half of the 20th century. In the past decades, a regular short distance dispersal eastward and westward became predominant.

## Phylogeography from full-length sequences and ORFs of 50 representative isolates

The spatio-temporal dispersal of RYMV in East Africa was reconstructed from sequences of 50 isolates representative of the genetic and geographic diversity of the virus in East Africa: full-length sequences (4468 nt, but excluding recombinant regions), sequences of ORF2a (including protease and VPg, 459 codons), ORF2b (including polymerase; 540 codons) and ORF4 (capsid protein, 240 codons). The differences in nucleotide diversity and in the selection pressure among the ORFs are given in the S5 Table. No recombination event was detected within the ORF2a, ORF2b and ORF4.

To make up for the weak temporal signal of these small datasets (50 sequences), a temporal prior was applied to the root of each tree by using the TMRCA of the EA335 tree, following a normal distribution with a mean of 170 years and a standard deviation of 25 years. Regardless the differences in sequence length, diversity and selection pressure between the ORFs (S5 Table), the phylogeographic patterns were similar (Fig 3), and consistent with that reconstructed from the 335 sequences of the coat protein gene: (1) an emergence in the Eastern Arc Mountains, (2) a rapid spread to the adjacent large rice growing valley of Kilombero, (3–4) a short range dispersal towards Morogoro and a long range dissemination northward towards Lake Victoria, (5) spread towards the south of Lake Malawi, (6) spread towards northern Ethiopia, (7) generalized dissemination throughout East Africa.

## Phylogeography of the three main lineages

The initial diversification of RYMV in East Africa resulted in the emergence of the three lineages S4, S5 and S6 in the mid-19th century within or near the Kilombero valley. The phylogeography of the three lineages is markedly different (Fig 4A and 4B). The S4 lineage had the greatest epidemic success. It was found south of Lake Victoria in the second half of the 19th century (S4lv), and then circulated around Lake Victoria (S4ug) before dispersing northward into Ethiopia (S4et), then southward to Malawi (S4lm) and finally westward to the Republic of Congo, Rwanda and Burundi. The S6 lineage remained confined to the Kilombero valley and in the Morogoro region for several decades. In the last decades, it spread towards eastern Tanzania, south-west Kenya and to the islands of Zanzibar and Pemba. The S7 strain split early from the S6 lineage and was recovered recently in southern Malawi. The S5 lineage was found exclusively in the Kilombero valley and in the Morogoro region. The migration curve (Fig 4C) illustrates the differences in velocity among the lineages, with an early and rapid spread of the S4 lineage, a late but rapid spread of the S6 lineage (except the early spread of the S7 strain), and a lack of dispersal for S5. Interestingly, there was a slowdown in the dispersal rates the past decades within most strains of the S4 and S6 lineages.

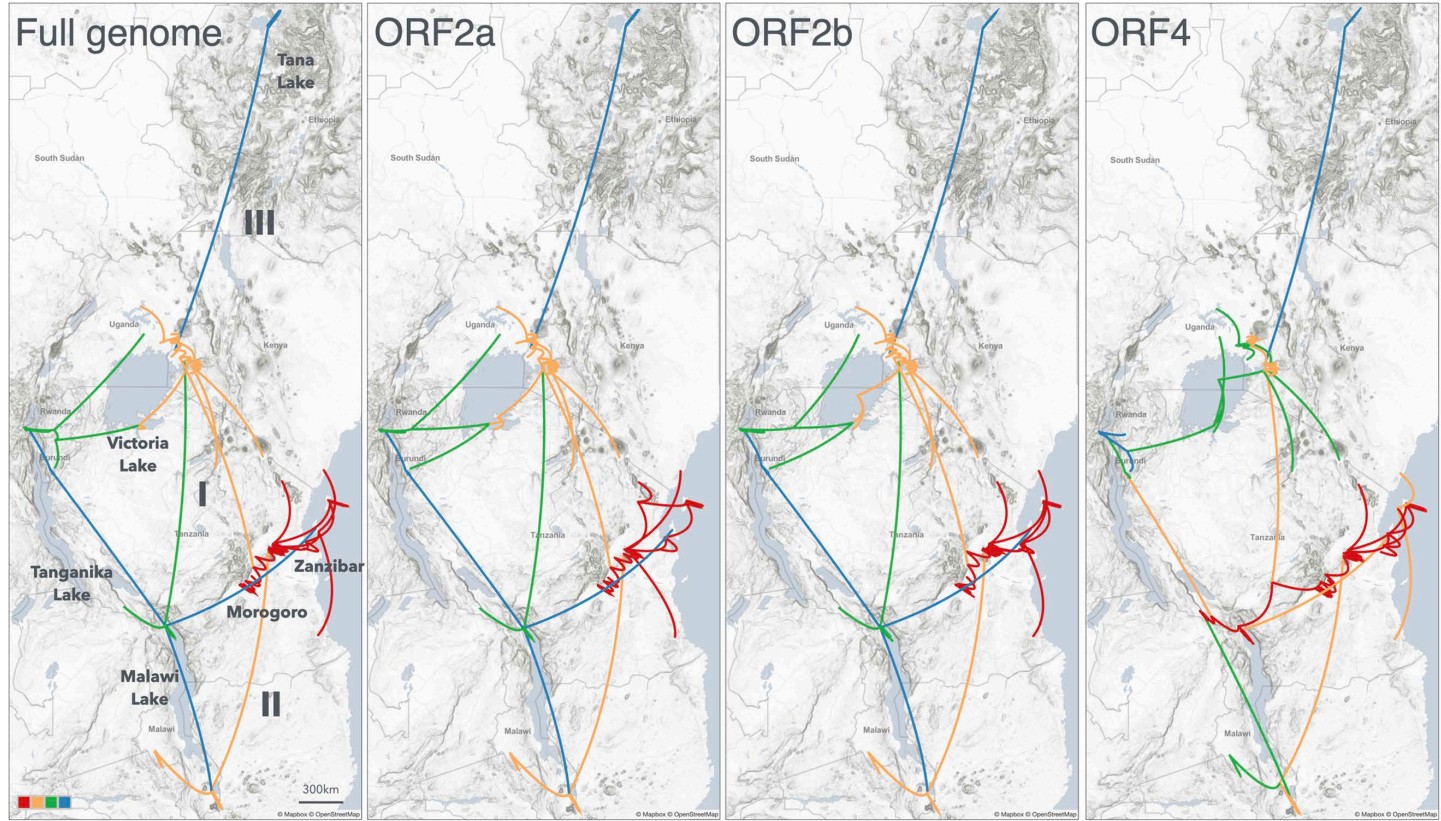

**Fig 3. Phylogeographic reconstructions of RYMV in East Africa from the different ORFs.** Continuous phylogeographic reconstructions of the spatiotemporal dispersal of RYMV lineages in East Africa from 1850 to 2020 based on the non-recombinant full-length sequences, ORF2a and ORF2b and ORF4 of 50 representative isolates of the genetic and geographic diversity. The recombinant regions were removed from the full-length sequences (see Materials and Methods). The phylogeographic scenario is displayed with temporal gradients linked to line color and curvature. Prints use map data from Mapbox and OpenStreetMap and their data sources. To learn more, visit https://www.mapbox.com/about/maps/ and http://www.openstreetmap.org/copyright.

Additional surveys were conducted from 2013 to 2017 at the Kilombero Rice Plantation (KPL) rice intensification station, located within the Kilombero Valley, which covers 5 000 ha of rice. A total of 75 isolates was collected in rice fields under intensifying schemes and in surrounding traditional rice fields. The coat protein gene of these isolates was sequenced. Almost all of the isolates belong to strain S6b already found in this region. Only one isolate of strain S6a, one S5 isolate and four S4lm isolates were collected. The low number of isolates, other than of the S6b strain, shows that the introduction of strains in the Kilomobero rice valley was restricted, even in the present time, reflecting its epidemiological isolation.

## East African origin of the introduction of RYMV into West Africa and into Madagascar

The main trends of dispersal of RYMV throughout East Africa, West Africa, Madagascar were captured within a single RRW model. This underlines the flexibility of the RRW models to reconstruct spatio-temporal dispersal when long distance jumps, even transcontinental ones, occurred. The phylogeny of RYMV was reconstructed from the full-length sequences of 101 isolates representative of the genetic and geographic diversity of the virus in Africa: 50 isolates from East Africa, 45 from West Africa and six sequences from Madagascar. A major split in the phylogenetic network separates East Africa and

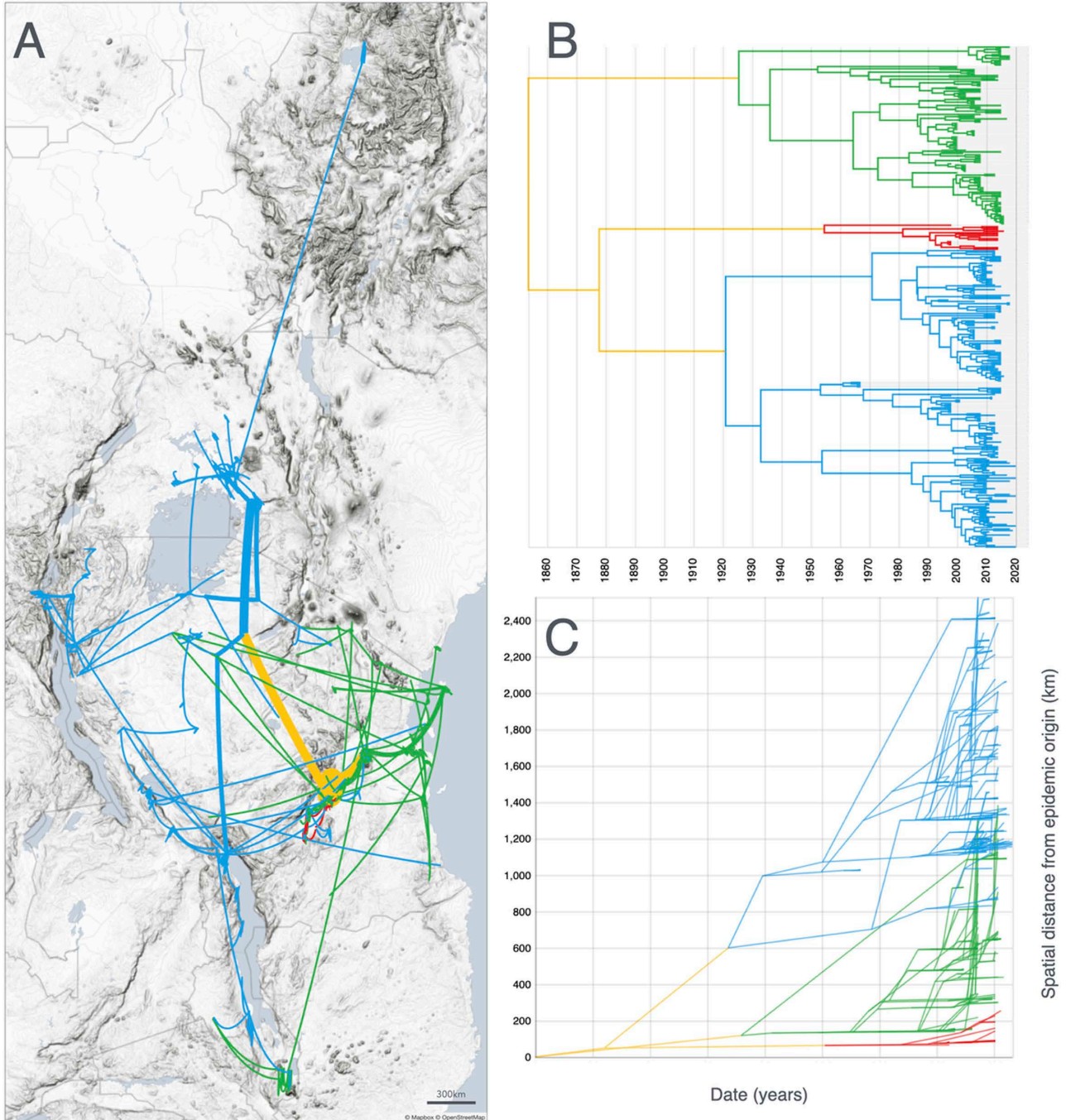

**Fig 4. Phylogeography of the three main lineages of RYMV in East Africa. (A)** Continuous phylogeographic reconstruction of the spatiotemporal dispersal of the three main lineages of RYMV in East Africa from 1850 to 2020 based on the coat protein gene sequences of 355 isolates collected from 1966 to 2019. The three lineages were distinguished by their colors: S4 (blue), S6 (green), S5 (red). Note that the S5 lineage, confined to the Kilombero valley and adjacent Morogoro region, is nearly masked by the S6 lineage that has a broader distribution. **(B)** Maximum credibility tree with time line reconstructed from the dataset and used to generate the spatio-temporal dispersal (same color code for each lineage). **(C)** Migration curves over time (preorder transversal of the phylogenetic tree) for each parent/child node pair: X-axis = cumulative branch length (as dates in years), Y-axis = cumulative great-circle distances given latitudes and longitudes. Prints use map data from Mapbox and OpenStreetMap and their data sources. To learn more, visit https://www.mapbox.com/about/maps/ and http://www.openstreetmap.org/copyright.

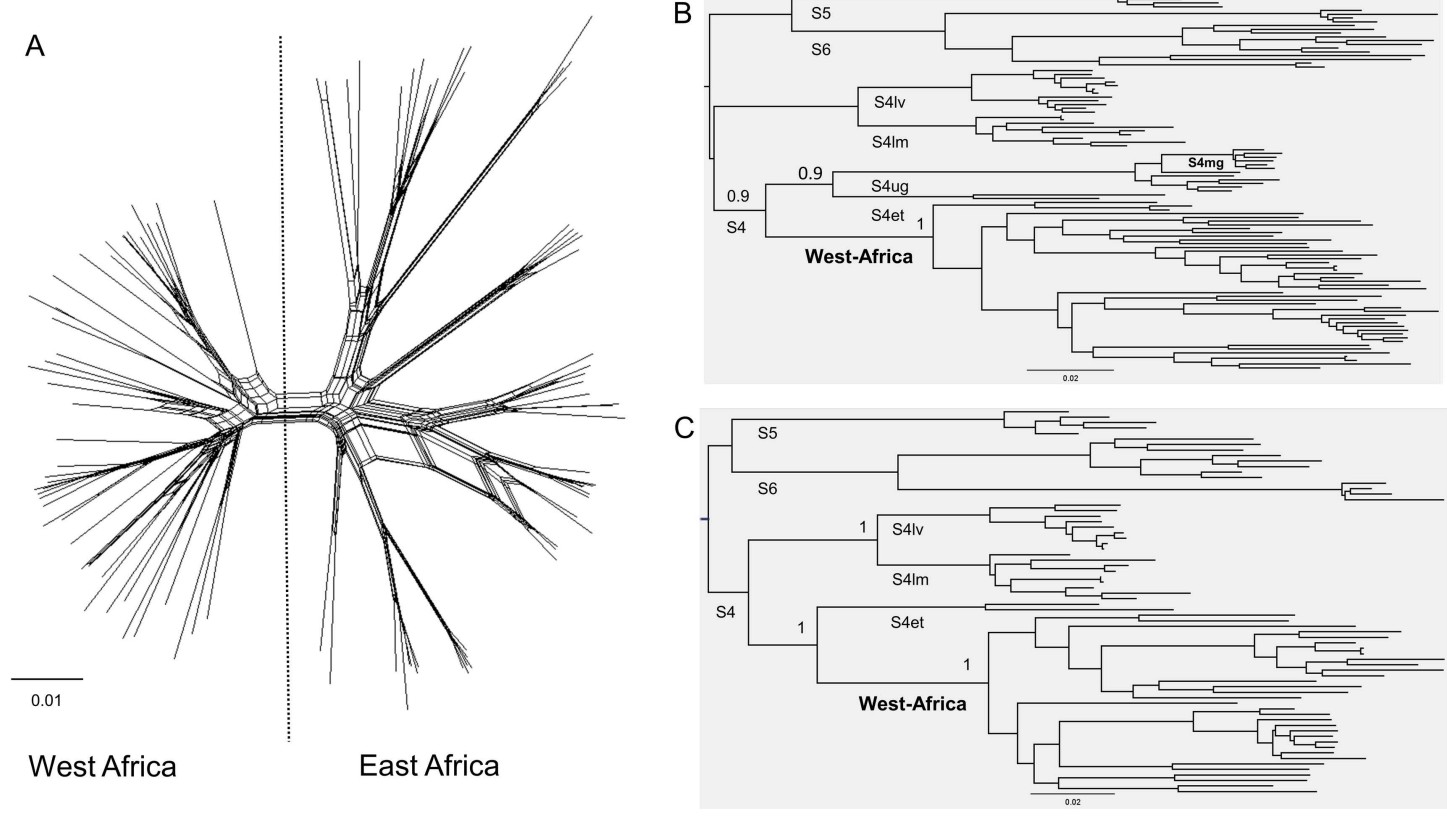

**Fig 5. NeighborNet and phylogeny of RYMV in Africa reconstructed from the full genome.** NeighborNet phylogenetic network of the 101 fully sequenced isolates of RYMV from East Africa and West Africa under a HKY85 distance model **(A)**. Maximum likelihood phylogeny tree reconstructed under a HKY85 model and mid-point rooting of the recombination-free full sequences obtained after removing the recombinant regions **(B)**, and after removing the recombinant sequences **(C)**.

Madagascar isolates from West Africa ones (Fig 5A). The phylogeny is characterized by the nested position of the West African lineage within the East African lineage comprising the S4ug, S4mg and S4et strains (Fig 5B). The S4mg strain in Madagascar resulted from a recombination between the S4ug strain and an unidentified minor parent [14]. The relationship between the West-African lineage and the S4et strain of this lineage persist after excluding the S4mg and S4ug sequences (Fig 5C). This confirms the earlier assumptions, made on a restricted dataset, that RYMV in West Africa and in Madagascar originated in East Africa [14,15], and brings information on the source lineage of RYMV in West Africa.

We detailed the East African origin of the introduction of RYMV into West Africa and into Madagascar. Convergence of the analyses with the full dataset (AE335, WA335 and Mg94 dataset, i.e., a total of 784 coat protein gene sequences) was difficult to reach. Therefore, 50 isolates representative of the diversity of the virus in West Africa and 15 isolates representative of Madagascar were selected using PDA analyzer [43]. Then we reconstructed the phylogeography of the collection of the coat protein gene of 335 isolates from East Africa, the 50 representative isolates from West Africa and the 15 representative isolates from Madagascar, i.e., a total of 400 isolates. The phylogeographic model was that defined for the analysis of the EA335 sequence dataset (see Materials and Methods). The split between the East and the West African clades occurred in ca. 1874 (95% HDP 1843–1912) at the south of Lake Victoria (Fig 5). The introduction of RYMV in the Niger Inner Delta in West Africa occurred in ca. 1897 (95% HDP 1870–1921). The subsequent pattern of dispersion throughout West Africa and toward Central Africa mirrored the findings from comprehensive surveys conducted in West

**Table 2. Dispersal metrics of RYMV in East Africa, West Africa, Madagascar and the three regions combined[1].**

| Dataset | Weighted diffusion coefficient (km²/year) | IBD signal (rP)[2] | Number of samples | Study area (km²)[3] | Reference |
|---|---|---|---|---|---|
| East Africa | 1081 [897, 1257] | 0.37 [0.30, 0.45] | 335 | ~2 103 000 | this study |
| Madagascar | 1246 [622, 1876] | 0.35 [0.28, 0.44] | 94 | ~446 000 | [14] |
| West Africa | 1603 [1159, 2166] | 0.56 [0.45, 0.64] | 210 | ~2 979 000 | [74] |
| West Africa and the Niger Valley | 1258 [953, 1585] | 0.52 [0.44, 0.62] | 261 | ~3 087 000 | [44] |
| East Africa, West Africa and Madagascar combined | 2718 [2259, 3379] | 0.39 [0.24, 0.60] | 400 | ~12 522 000 | this study |

[1]For each dataset and metric, we report both the posterior median estimate and the 95% highest posterior density (HPD) interval.

[2]The isolation-by-distance (IBD) signal has been estimated by the Pearson correlation coefficient (rP) between the patristic and log-transformed great-circle geographic distances computed for each pair of virus samples.

[3]The extent of the study areas were approximated by the inland area of the minimum convex hull polygon surrounding all the sampling.

Africa [44,45]. In Madagascar, RYMV was introduced in ca. 1980 (95% HPD 1975–1986) from the northeast of Lake Victoria. The introduction of RYMV in the northwest of Madagascar followed by the spread of the virus southward of the country is a scenario similar to that obtained from the comprehensive dataset [14].

## Comparative dispersal statistics of RYMV

For RYMV, variation in isolation-by-distance (IBD) signals among the three regions (East Africa, West Africa and Madagascar) was apparent (from 0.35 to 0.56) (Table 2). The IBD index was higher in West Africa (0.52 and 0.56 for the two datasets) than in East Africa (0.37) and in Madagascar (0.35) (Table 2). In East Africa, the recurrent long distance human-mediated dispersal of RYMV in a wide range of directions revealed by this study is consistent with a low IBD signal. The higher IBD signal estimated in West Africa may result from the dominant eastward long-distance transmission events establishing more local transmission chains [44,45,46]. The relatively low IBD signal estimated in Madagascar suggested that long-distance human-mediated dispersal, which blurred the relationship between geographical and genetic distances, is also frequent in a country where rice is widely grown. Diffusion coefficients in the three regions ranged from 1100 to 1600 km² per year with overlapping 95% HPD intervals. This indicates a similar dispersal capacity of the virus across the three regions despite differences in IBD signal values, spatial scales, rice history and cultivation, and possibly also in dispersal modes (Fig 6).

The diffusion coefficient of RYMV for East Africa, West Africa and Madagascar combined (covering ca. 12 million km²) was 2700 km²/year, was twice that of each of three regions reflecting the inclusion of the long-distance dispersal from East Africa towards West Africa and towards Madagascar (Table 2).

## Discussion

The historical data provided circumstantial evidence that support, explain and complement the interpretation of the phylogeographic reconstructions. The seven periods identified in the phylogeographic reconstructions of RYVM in East Africa are interpreted as followed on the basis of the history of rice since the beginning of the 19th century.

(1) The postulated emergence of RYMV in the Eastern Arc Mountains [47] was confirmed and characterized in more details. The emergence of RYMV dates back to the mid-19th century. It took place near the Great Ruaha Escarpment of the Udzungwa Mountains adjacent to the Kilombero valley and south to Morogoro. Consistently, the insertion-deletion polymorphism originated in isolates of this region. The original host(s) of RYMV were located within the Eastern Afromontane biodiversity hotspot [15,47]. Slash and burn rice cultivation practiced in the

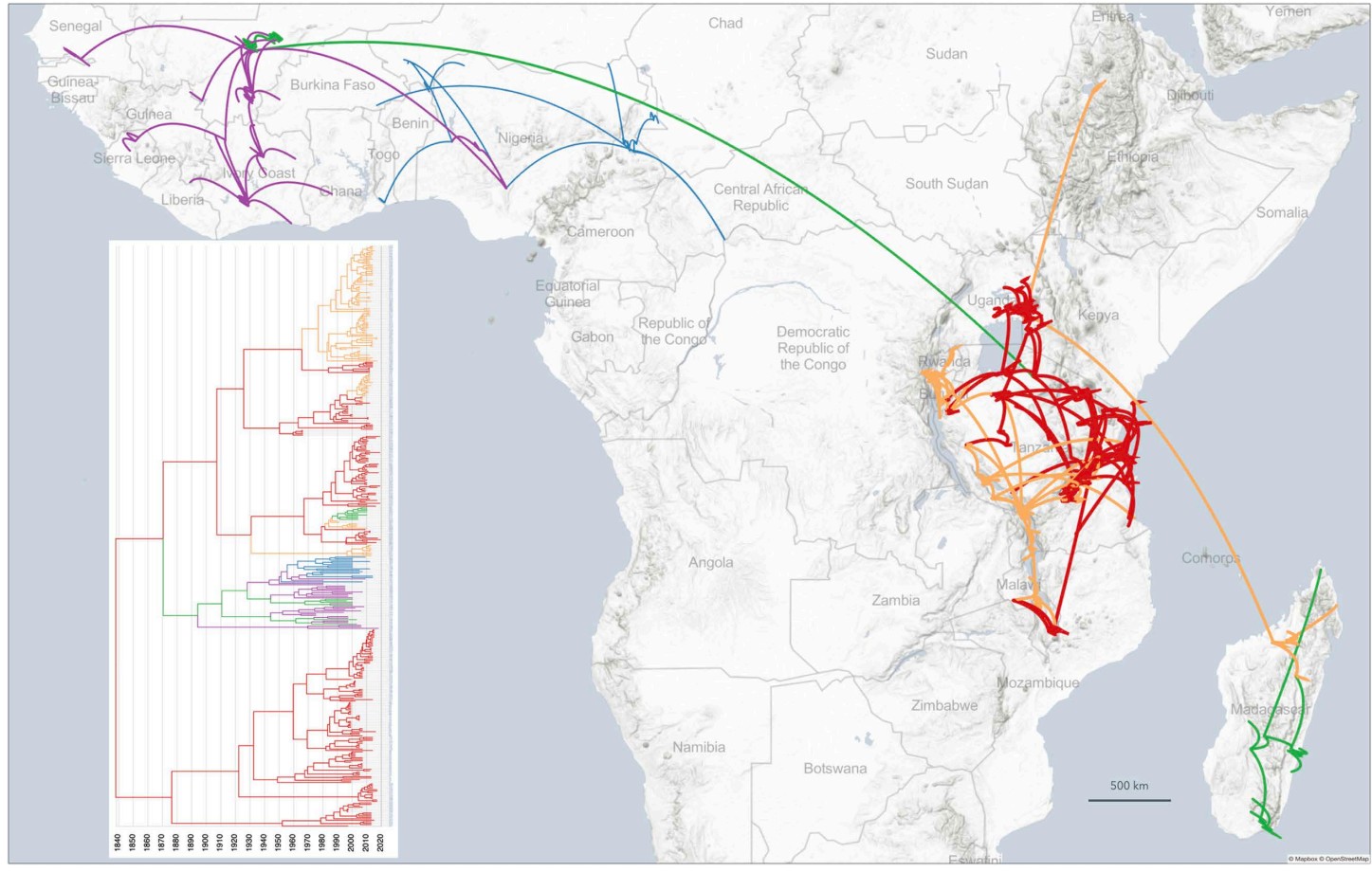

**Fig 6. Dispersion of RYMV from East Africa to West Africa and Madagascar.** Dispersion of RYMV from East Africa (red and orange) to West Africa (purple and blue) and to Madagascar (pale green) between 1850 and 2020 reconstructed from the ORF4 sequences of 335 isolates from East Africa, 50 isolates from West Africa and 15 isolates from Madagascar. Temporal gradients are linked to line color. Inset: Maximum credibility tree reconstructed from the dataset and used to generate the spatio-temporal dispersal (same color code). Prints use map data from Mapbox and OpenStreetMap and their data sources. To learn more, visit https://www.mapbox.com/about/maps/ and http://www.openstreetmap.org/copyright.

mid-1800s near the Great Ruaha Escarpment of the Udzungwa Mountains [29,33,48,49] resulted in spillovers to cultivated rice, presumably – given RYMV host range – from the primary hosts of the *Oryzeae* tribe of the subfamily *Oryzoideae* and of the *Eragrostidae* tribe of the subfamily *Chloridoideae* within the grass family *Poaceae*. Primary hosts may also include the finger millet (*Eleusine corocana*) of the subfamily *Chloridoideae*, a local host of RYMV [5,6], widely cultivated through slash and burn cultivation in these mountains during the 19th century [48]. This led to introductions of RYMV in cultivated rice. The basal split between the S4, S5 and S6 lineages and the insertion-deletion polymorphism suggest (at least) three independent spillovers from the wild host(s) to cultivated rice. Strains specific poaceae species such as *Dactyloctenium aegyptium* have been identified [50]. Specific mutations of the S7 strain occurred at positions of the VPg under positive selection and experimentally associated to host range [51,52]. Further studies are necessary to understand the role of the genetic changes, if any, associated to host jumps and rice spillover.

(2) Within a few decades, RYMV spread to the large adjacent Kilombero Valley where rice had been widely cultivated since the beginning of the 19[th] century [29]. This is consistent with the movements of rice-growing populations between the Udzungwa Mountains and the Kilombero Valley [29].

(3) Between the second half of the 19[th] century and the beginning of the 20[th] century, RYMV was introduced in the Morogoro region and spread to the south of Lake Victoria 400 km northward. Its dispersal followed the introduction of rice into the interior of the continent along the caravan routes linking the coastal strip to Lake Victoria [53, 54, 55]. There are indications that the caravan routes crossed the Morogoro region and the Kilombero valley [33,56]. Interestingly, the migration pathways of turnip mosaic virus (TuMV), a seed transmitted potyvirus, retraced some of the historical trade arteries of the Silk Road [57]. So, both RYMV and TuMV illustrate the link between plant virus dispersal and trade routes.

(4) In the following decades, up to the middle of the 20[th] century, RYMV dispersed at the east of Lake Victoria where rice had been introduced at the end of the 19[th] century, along the caravan roads [56] or by boat [58,59]. Interestingly, the first field report of RYMV in East Africa was in 1966 near Kisumu also at the east of Lake Victoria [5].

(5) At the beginning of the 20[th] century, RYMV was introduced from the Kilombero valley to the south of Lake Malawi. This movement was parallel and concomitant to the movement of military troops at the end of the First World War [60,61]. Feeding the troops was a major issue [62] and rice was the main staple food for soldiers. The Kilombero valley, the Rufiji basin and the Morogoro region were the main rice production areas. Accordingly, RYMV may have been dispersed through the transport of contaminated seeds during the First World War. An alternative hypothesis, linking virus dispersal to the Kilwa-Nyasa south caravan route, from Kiwa 300 km south of Zanzibar to the south of Lake Malawi (formerly named Lake Nyasa), is less likely. First, there is no firm indication that rice had been introduced at the south of Lake Malawi along this caravan route [63,64]. Second, the peak of activity along the Kilwa-Nyasa south caravan route occurred in the second half of the 19[th] century [63,64] whereas the introduction of RYMV at the south of Lake Malawi occurred at the beginning of the 20[th] century.

(6) In the second half of the 20[th] century, RYMV was introduced into northern Ethiopia from the north of Lake Victoria. Large rice intensification schemes were established in the 1960s around Lake Victoria [5] and in the 1980s around Lake Tana in northern Ethiopia [65]. Seed exchanges between these intensification schemes were frequent and may have introduced RYMV in northern Ethiopia. The widespread cultivation of teff (*Eragrostis tef*), a host of RYMV [5], in the Ethiopian highlands may have facilitated the dissemination.

(7) RYMV then spread eastward and northward from the Morogoro region (strain S6a) and from the Kilombero Valley (strain S6b), and southward and westward from the Lake Victoria region (strains S4). RYMV finally became widespread throughout East Africa. This spread accompanied the expansion of rice cultivation of the last decades of the 20th century and of the beginning of the 21st century. The slowdown in the dispersal rates of several strains of the S4 and S6 lineages over the past decades reflected a shift from an expansion phase to an endemic phase [34,35] in RYMV epidemiology. This may be due to changes in the trade and market patterns [66] with a gradual change from long distance to local seed transport as rice became widely cultivated.

The phylogeography of RYMV throughout East Africa unveiled the role of rice transport in the long-distance movement of the virus. Similarly, the introduction of RYMV in West Africa may result from seed movements. Seed exchanges in Africa between agricultural stations, experimental farms, botanical and missionary gardens, even far apart, were frequent at the end of the 19[th] century [67,68]. The Niger Inner Delta was a major destination for seeds from a wide range of origins, including Madagascar, because of rice intensification in this region since the end of the 19[th] century [69,70]. The origin of RYMV introduced in West Africa was located at the south of Victoria Lake at the end of the 19[th] century, in Tanzania, which was under German occupation at that time. Early contacts between German scientists and agricultural stations

of the Niger Inner Delta are documented [68]. RYMV may have been introduced directly from this region, or indirectly through the Amani research station which was the main agricultural station in East Africa [71]. Our results also confirm the Ugandan origin of the Madagascar strain [14]. Increased seed exchanges between Madagascar and East African countries have occurred during the recent decades [73]. Transport of contaminated rice seeds may also account for this introduction.

Given a host range that is restricted to rice and closely related poaceae species, a direct connection between RYMV epidemiology and rice ecology was anticipated [10]. However, elucidating the precise nature of this relationship has proven challenging. Linear relationships between the intensity of rice cultivation and the transition frequency of RYMV lineages in West-Africa have been proposed [15], as well as between rice cultivation intensity and virus population size in the irrigated valley of Niger [44]. Nevertheless, further statistical analyses failed to provide supports for such associations between the dispersal dynamic of viral lineages or virus population size and rice-cultivated areas [44,74]. Factors that make the relationship between RYMV epidemiology and its hosts more complex than initially proposed in early modelling attempts have been explored [14,44]. Interestingly, in East Africa, the transfer of rice seeds, rather than rice cultivation was critical in RYMV dispersal. Virus spread in East Africa was more closely associated with human-mediated long-distance transport of rice seeds than with the short-range vector-borne movements typically linked to rice cultivation.

RYMV dispersal through the transport of contaminated seeds shows the importance of this neglected virus transmission pathway in plant virus epidemiology [1]. This study also unveils the wide range of human activities, some of them unsuspected, involved in man-mediated movements of a plant virus. The diffusion coefficient of RYMV was higher than that of several animal viruses including rabies in various hosts, even when occasional rapid long-distance dispersal events driven by human-mediated movements occurred [35]. Yet, the diffusion coefficient of RYMV was lower by an order of magnitude than that of animal viruses with long distance dispersal through animal trade or through migratory birds [35].

Maize streak virus A (MSV-A) causes another major cereal disease in Africa [75]. It is transmitted by leafhoppers and not by seeds. Interestingly, introductions of MSV-A from Uganda to Madagascar [73] and from Kenya to Ethiopia [76] have been reported. These movements, parallels to those of RYMV, were also attributed to human-mediated introductions [73,76]. It shows that the spread of RYMV and MSV-A, two major vector borne viruses, does not necessarily, or even primarily, depend on vector transmission.

An important pending question is why RYMV, given its wide dispersal capacities, is still restricted to Africa and does not have a worldwide distribution, unlike several sobemoviruses with similar biological properties. Long-distance movement of RYMV across the sea is possible but is quite a rare event as indicated by the unique successful introduction of the virus in West Africa. Rice was imported into America from West Africa in the 17th century and early 18th centuries, at least a century before the introduction of RYMV in West Africa. In the 19th century, rice exports from East Africa were limited and directed towards regions where rice was not cultivated [31]. In the 20th century, rice was imported into Africa rather than exported from this continent. Altogether, this could explain why RYMV remained restricted to Africa. However, the risk of accidental introduction of RYMV into other continents subsists, in particular through the transfer of seeds for agronomical breeding which we have found to be a decisive means of long distance dissemination of the virus in Africa. Incidentally, MSV confined so far to Africa has recently been reported in the Philippines [77], a possible illustration of such human-mediated trans-continental introductions of vector borne plant viruses.

## Materials and methods

### Datasets

A collection of 335 capsid protein gene sequences of East African isolates was built. The dataset consists of 240 sequences retrieved from NCBI (2024-06-01) supplemented by 95 sequences obtained in this study (S1 Table). The isolates originated from eight countries (Burundi, Republic of Congo, Ethiopia, Kenya, Malawi, Rwanda, Tanzania, Uganda). The sampling locations were distributed between latitudes 15.6° South and 11.9° North, and from longitudes 28.8° West to 39.8° East, ca.

3 000 kilometers in latitude x 1 200 km in longitude, i.e., a study area of ca. two million square kilometers. The isolates were sampled from 1966 to 2019 (54 years). A minor variant named S4ke recently identified in Kenya with a recombinant event in the capsid protein [78] was considered in this study (see below) but not included in this corpus.

The genome of RYMV is organized into five overlapping open reading frames. ORF1, located at the 5' end, encodes a protein involved in virus movement and gene silencing suppression. ORFx overlaps with the 5' end of ORF2a and is translated through a leaky scanning mechanism. ORF2a and ORF2b are overlapping ORFs that encode the central polyprotein. ORF2a encodes a serine protease and the VPg. ORF2b translated through a -1 ribosomal frameshift encodes the RNA-dependent RNA polymerase. ORF4 located at the 3' end encodes the coat protein. This ORF is expressed via a subgenomic RNA [3].

A collection of full-length sequences of 101 isolates from Africa was set. This dataset consisted of sequences of 50 isolates of East Africa and of the East of Central Africa (referred to as the East African dataset), 45 isolates of West Africa and of the West of Central Africa (referred to as the West African dataset) and six sequences of Madagascar. Among the 50 full-length sequences of the East African dataset, 23 were sequenced for this study and 27 were retrieved from NCBI (S2 Table). This sample is representative of the genetic and geographic diversity of RYMV in East Africa, including the S4ke variant from western Uganda. No recombination signal was detected within the ORF2a and ORF2b genes.

In complement, a collection of 335 capsid protein gene sequences from West African isolates, was considered to estimate the substitution rate of RYMV (S3 Table). It consists of sequences retrieved from NCBI (2024-06-01), 261 sequences used in a study of the phylogeography in West Africa [44] complemented by 46 sequences of Ghana [45] and 28 sequences of Burkina Faso [46]. The 28 sequences from Burkina Faso were chosen from the 144 available on NCBI using the PDA software [43] in order to maximize the genetic diversity of the sample. In total, the samples were collected in 15 countries (Benin, Burkina Faso, Cameroon, Central African Republic, Chad, Gambia, Ghana, Guinea, Ivory Coast, Mali, Niger, Nigeria, Senegal, Sierra Leone, Togo). They were sampled between 1975 and 2021 (47 years).

## Phylogenetic reconstructions

Sequences were aligned using CLUSTAL X with default parameters [79]. Maximum-likelihood (ML) phylogenetic trees were inferred using the PHYML-3.1 algorithm implemented in the SEAVIEW version 4.7 software [80] under a HKY85 substitution model. The ML trees were rooted at the point in the tree that minimizes the residual mean square of the root-to-tip distances. The neighborNet phylogenetic network of the 101 fully sequenced isolates was inferred [81] under an HKY85 distance model as implemented in SplitsTree version 4. The neighborNet phylogenetic network illustrates the genetic relationships between sequences taking into account the conflicting phylogenetic signals that are possibly due to recombination events. The full-length sequence alignment was screened for recombination signals using RDP5 version [82]. The default settings were used for each of the seven recombination detection algorithms that RDP incorporates, as was a Bonferroni corrected P-value cut-off of 0.05. Only recombination events detected by more than four of the seven methods implemented in RDP were considered. The phylogeny was reconstructed from recombinant-free alignments. This was obtained by removing either the recombinant sequences or the recombinant regions. In the latter case, it was checked that further recombination was excluded through the pairwise homoplasy (PHI) test [83] implemented in SplitsTree version 4 [81]. Total, synonymous, and non-synonymous nucleotide diversity of ORF2a, ORF2b and ORF4 were assessed by DnaSP version 6 [84]. The ORF1, too short, too variable and with a low phylogenetic signal [85], was not considered.

## Bayesian evolutionary inferences

RYMV is a measurably evolving population, yet its overall temporal signal, depending on the dataset, is weak to moderate [14,15]. The temporal signal of each data set was assessed through tip cluster-randomization tests in BEAST [86] and by BETS [37] under a Bayesian statistical framework, and by TreeTime under a likelihood framework [87]. We subsequently reconstructed time-calibrated phylogenies using a Bayesian statistical framework implemented in BEAST version 1.10.4

software [88]. BEAST uses Markov Chain Monte Carlo integration to average over all plausible evolutionary histories for the data, as reflected by the posterior probability. All analyses were performed using the BEAGLE library to enhance computation speed [89,90]. We specified an HKY85 substitution model with a discretized gamma distribution (four categories) to model rate heterogeneity across sites. To accommodate variation of substitution rates across lineages, an uncorrelated relaxed molecular clock that models the branch rate variation according to a lognormal distribution was specified [91]. The flexible nonparametric demographic skygrid prior was selected to accommodate for variation in the rate of coalescence through time [92]. Stationarity and mixing (e.g., based on effective sample sizes > 200 for the continuous parameters) were assessed using Tracer version 1.7 [93]. Consensus trees deriving from the phylogenies sampled during the MCMC analysis were obtained using TreeAnnotator version 1.10.4 [88].

To study the geographical spread of RYMV in continuous space in East Africa and to quantify its tempo and dispersal, we fitted a continuous phylogenetic diffusion model to the East African dataset, modelling the changes in coordinates (latitude and longitude) along each branch in the evolutionary history as a bivariate normal random deviate [22]. As a more realistic alternative to homogeneous Brownian motion, we adopted a "Cauchy" RRW extension. It models variation in dispersal rates across branches by independently drawing branch-specific rate scalers of the RRW precision matrix from a gamma distribution with shape and scale equal to 1/2 to relax the assumption of a constant spatial diffusion rate throughout the whole tree [22]. The RRW that was used in this study is equivalent to a Cauchy process on a tree with rescaled branch lengths, which is a pure jump process allowing for long-distance dispersion events [72]. Bayesian inference using continuous diffusion models yields a posterior distribution of the phylogenetic trees, each having ancestral nodes annotated with location estimates. The EA335 dataset was used to reconstruct the dispersal of RYMV throughout East Africa. A spatial jitter of 0.1° in latitude and in longitude (i.e., ca 10 km) was applied to the locations of the isolates. This degree of noise for identical coordinates was needed to avoid improper posteriors under the RRW model and associated inference problems [22,94]. Evolaps version 3 was used to visualize the continuous phylogeographic reconstruction of the dispersal history of RYMV lineages [95,96]. The phylogeography of RYMV in East Africa was reconstructed from the coat protein gene sequences of the full dataset (335 isolates), from the full genome and different ORFs of a subsample of the 50 isolates representative of the genetic and geographic diversity of the virus.

### Comparative dispersal statistics of RYMV

The phylogeography of RYMV has been reconstructed in East Africa (this study), in West Africa [44] and in Madagascar [14]. The dispersal capacity and dynamics in the three regions was compared through the weighted diffusion coefficient (km²/year) [15] and the isolation-by-distance (IBD) signal metric, two metrics identified as robust (to sampling intensity in particular) and measuring complementary aspects of the overall dispersal pattern [35]. The diffusion coefficient is an estimate of the dispersal capacity of the lineages. The IBD signal metric quantifies how the dispersal is spatially structured, i.e., the tendency of phylogenetically closely related tip nodes to be sampled from geographically close locations. An IBD signal was apparent through the relationships between geographic and genetic pairwise distances in a restricted dataset of sequences from East and West Africa [97]. The diffusion coefficient of RYMV virus over the three regions combined was also estimated. Then, the dispersal capacity of this plant virus was compared to that of a range of animal viruses in various host species [35].

### Supporting information

**S1 Table. Name and accession number of the 335 isolates from East Africa.**
(XLSX)

**S2 Table. Name and accession number of the 101 fully sequenced isolates from West Africa, East Africa and Madagascar.**
(XLSX)

**S3 Table. Name and accession number of the 335 isolates from West Africa.**
(XLSX)

**S4 Table. Name and accession number of the 94 isolates of Madagascar.**
(XLSX)

**S5 Table. Length, diversity and selection pressure of the ORF of Rice yellow mottle virus.**
(XLSX)

## Acknowledgments

This work benefited from insightful discussions with Dick Peters over the years on rice yellow mottle virus and on the importance of abiotic transmission in the epidemiology of plant viruses. Dick Peters passed away in November 2023.

We thank the IRD i-Trop HPC at IRD Montpellier for providing HPC resources that have contributed to some of the research results reported within this article. URL: http://bioinfo.ird.fr/.

## Author contributions

**Conceptualization:** Pauline Rocu, Simon Dellicour, Philippe Lemey, Erik Gilbert, Marie-José Dugué, François Chevenet, Paul Bastide, Stéphane Guindon, Denis Fargette, Eugénie Hébrard.

**Data curation:** Agnès Pinel-Galzi.

**Investigation:** Innocent Ndikumana, Geoffrey Onaga, Agnès Pinel-Galzi, Pauline Rocu, Judith Hubert, Hassan Karakacha Wéré, Antony Adego, Mariam Nyongesa Wéré, Simon Dellicour, François Chevenet, Paul Bastide, Denis Fargette.

**Project administration:** Denis Fargette, Eugénie Hébrard.

**Resources:** Innocent Ndikumana, Geoffrey Onaga, Judith Hubert, Hassan Karakacha Wéré, Antony Adego, Mariam Nyongesa Wéré.

**Software:** Maxime Hebrard, François Chevenet.

**Writing – original draft:** Denis Fargette.

**Writing – review & editing:** Innocent Ndikumana, Geoffrey Onaga, Agnès Pinel-Galzi, Pauline Rocu, Judith Hubert, Hassan Karakacha Wéré, Antony Adego, Mariam Nyongesa Wéré, Nils Poulicard, Maxime Hebrard, Simon Dellicour, Philippe Lemey, Erik Gilbert, Marie-José Dugué, François Chevenet, Paul Bastide, Stéphane Guindon, Denis Fargette, Eugénie Hébrard.

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
