## [Decision Letter · Decision Letter 0]

26 Nov 2024

PPATHOGENS-D-24-01977Grains, trade and war in the multimodal transmission of Rice yellow mottle virus: an historical and phylogeographical retrospectivePLOS Pathogens Dear Dr. Hebrard, Thank you for submitting your manuscript to PLOS Pathogens. After careful consideration, we feel that it has merit but does not fully meet PLOS Pathogens's publication criteria as it currently stands. Therefore, we invite you to submit a revised version of the manuscript that addresses the points raised during the review process. Please submit your revised manuscript within 30 days Jan 25 2025 11:59PM. If you will need more time than this to complete your revisions, please reply to this message or contact the journal office at plospathogens@plos.org. Please include the following items when submitting your revised manuscript: * A rebuttal letter that responds to each point raised by the editor and reviewer(s). You should upload this letter as a separate file labeled 'Response to Reviewers '. This file does not need to include responses to any formatting updates and technical items listed in the 'Journal Requirements' section below. * A marked-up copy of your manuscript that highlights changes made to the original version. You should upload this as a separate file labeled 'Revised Manuscript with Track Changes '. * An unmarked version of your revised paper without tracked changes. You should upload this as a separate file labeled 'Manuscript '. If you would like to make changes to your financial disclosure, competing interests statement, or data availability statement, please make these updates within the submission form at the time of resubmission. Guidelines for resubmitting your figure files are available below the reviewer comments at the end of this letter. We look forward to receiving your revised manuscript. Kind regards,Carolyn M. MalmstromGuest EditorPLOS Pathogens Ronald SwanstromSection EditorPLOS Pathogens Michael Malim

Editor-in-Chief

PLOS Pathogens

orcid.org/0000-0002-7699-2064 **Additional Editor Comments :** Thank you for the submission of this interesting manuscript. I am returning it with three reviews. The three reviewers all find the study to be of notable interest to a general audience and describe it as "compelling" and "a major advance". I concur with these assessments. There are no major concerns with the analytical methods that would require additional work but a few points to discuss further. Importantly, all three reviewers indicate that the manuscript would be significantly stronger with improvements to the text and figures.

Please be attentive to all of the comments from the reviewers. It will be especially important to address the following points about figures from reviewer 1 and myself and about the text as summarized from reviewers 1-3. These are improvements that will strengthen your paper and clarify important points for the readers.

Figure points

1. Can figures 1 & 2 be annotated and/or combined in some way, as reviewer 1 suggests, to increase their value to the reader?

2. Are all of figures 3-6 essential? Can they be annotated/edited to support better comprehension? Is it possible to specify directionality to the lines as arrows? Please test the revised figures with a general scientific audience before re-submission.

3. Reviewer 1 found figures 7 and 9 to be most helpful, and I appreciated figure 8. Please consider whether these also can be "tuned up" further for maximal comprehension.

Text points

Introduction:

--The first paragraph provides essential information for the final conclusions but on first read may not appear fully connected to the rest of the text. Can you please provide a little more context and connection here?

Discussion:

--A common theme is a need to tighten the Discussion text (reduce repetition) and give it more narrative shape.

--In doing so, please address point 6 from reviewer 2 about the historical information, and points 1 and 2 that reviewer 3 raised under major issues (uncertainty about origins/movement, lack of temporal signal). Reviewer 3's points might be alternatively addressed in methods. Some of the discussion requested is present in the text but may need further highlighting so that readers more readily find it.

--Please further consider reviewer 1's suggestion of a timeline figure to illustrate the seven periods you have identified.

Finally, please consider the minor points presented by all three reviewers and respond as you see fit to best polish the manuscript quality. **Journal Requirements:**

2) We noticed that you used the phrases 'data not shown' and 'unpublished results' in the manuscript. We do not allow these references, as the PLOS data access policy requires that all data be either published with the manuscript or made available in a publicly accessible database. Please amend the supplementary material to include the referenced data or remove the references.

Potential Copyright Issues:

i) Figures 3, 4, 5, 6, 7a, and 9. Please (a) provide a direct link to the base layer of the map (i.e., the country or region border shape) and ensure this is also included in the figure legend; and (b) provide a link to the terms of use / license information for the base layer image or shapefile. We cannot publish proprietary or copyrighted maps (e.g. Google Maps, Mapquest) and the terms of use for your map base layer must be compatible with our CC BY 4.0 license.

5) We note that your Data Availability Statement is currently as follows: "All relevant data are within the manuscript and its Supporting Information files.". Please confirm at this time whether or not your submission contains all raw data required to replicate the results of your study. Authors must share the “minimal data set” for their submission. PLOS defines the minimal data set to consist of the data required to replicate all study findings reported in the article, as well as related metadata and methods (https://journals.plos.org/plosone/s/data-availability#loc-minimal-data-set-definition).

1) State the initials, alongside each funding source, of each author to receive each grant. For example: "This work was supported by the National Institutes of Health (####### to AM; ###### to CJ) and the National Science Foundation (###### to AM).".

7) Please ensure that the funders and grant numbers match between the Financial Disclosure field and the Funding Information tab in your submission form. Note that the funders must be provided in the same order in both places as well. This information "Agropolis Foundation (project no. 1504-004 E-SPACE), National Commission for Science, Technology and Innovation (NACOSTI) and Ministère de l’Europe et des Affaires Etrangères (MEAE)" is missing from the Funding Information tab.

Please indicate by return email the full and correct funding information for your study and confirm the order in which funding contributions should appear. Please be sure to indicate whether the funders played any role in the study design, data collection and analysis, decision to publish, or preparation of the manuscript.

**Reviewers' Comments:**Reviewer's Responses to Questions

**Part I - Summary**

Reviewer #1: This paper describes Rice yellow mottle virus transmission across several decades in East and West Africa and uses novel methods to track its evolution and spread. The authors present a very compelling story which will be of broad interest. While I would like to see this published, I have some comments mainly regarding the presentation of the results, which I think can be hugely improved.

Reviewer #2: This study reconstructs the history of RYMV dispersion and evolution in Africa. The study is based in a large data set of partial and full-length genome sequences that encompasses an unprecedented large spatial and temporal range, and on state of the art phylogenetic and phylogenomic procedures. The results of these analyses are then discussed in the light of circumstantial but confirmatory historical evidence. Throught the study of RYMV, the manuscript underscores the relevance human activities, from trade to war and scientific cooperation, may have in the dispersion of important plant pathogens. Although it is often assumed that human activities play major roles in the dispersal of plant pathogens, this is rarely shown. Thus, in my opinion this work is a major advance in understanding the long-term, long-range evolution of a pathogen, and the multiple factors that shape it.

Reviewer #3: Ndikumana et al. reconstruct the historical movement of rice yellow mottle virus (RYMV) through Africa using phylogeographic methods. This work adds nicely to the relatively small literature on the origins of major plant viruses. The narrative of the manuscript also nicely weaves phylogeographic reconstructions with what is known about the history of RYMV and rice production in the region. I do however have two concerns about the phylogeographic inferences.

**Part II – Major Issues: Key Experiments Required for Acceptance**

Reviewer #1: While the analysis is robust and I trust the methodology used to obtain these results, the way it is presented is the weakest part of this manuscript and doesn't do the paper justice. Most of the figures are very uninformative in their current form and require much more annotation to convey useful information. For example, Figures 1 and 2 could be combined into a single figure, and it would be useful to illustrate the congruence between them (i.e. using a tanglegram or similar). At the very least, some highlighting of clades and their position on the two trees would be helpful. As well, the + and - symbols could easily be replaced by showing the actual insertions and deletions for context. Figures 3, 4, 5, 6 and 8 seem like they are attempting to convey the same result but are unclear and uninformative. Figures 7 and 9 are the only ones that I feel provide enough useful information and are of adequate quality for a publication. The authors need to revise their presentation and have more consistency throughout to convey the story.

Reviewer #2: No major issue.

Reviewer #3: 1) The paper largely neglects uncertainty about the origins and movement of RYMV. This is particularly notable for the reconstructed history of events introduced on pages 10 and 11. It would be good to report measures of confidence/uncertainty surrounding the reconstructed ancestral locations and dispersal pathways in both the text and the figures. Similarly, phylogeographic reconstructions are well known to be highly susceptible to sampling biases. This raises the question of whether the history reconstructed here simply mirrors the geographic distribution of samples. Providing at least some discussion of these issues is important, especially for readers who may be less familiar with the limitations of these phylogeographic methods.

2) The lack of temporal signal in the molecular clock analysis is also perplexing. The root-to-tip regression shows surprisingly little correlation between sample dates and divergence for such a rapidly evolving virus. I'm therefore wondering if other factors may be obscuring the temporal signal here? For example is there unaccounted for recombination? Errors in the sampling dates? Or even possibly variation in the underlying clock rate (beyond what can be accounted for by the relaxed clock model) due some lineages evolving at higher rates? For a seed-borne virus it could be that some viral lineages essentially lie "dormant" while in seed and therefore do not accumulate substitutions at the rate expected for continuously replicating virus. Regardless of what's driving this, it might be worth inspecting the outliers for patterns that could explain this rate heterogeneity.

**Part III – Minor Issues: Editorial and Data Presentation Modifications**

Reviewer #1: Introduction - provides a good general overview of RYMV and its epidemiology. Some of the paragraphs seem to jump around a little, which affects the flow of information. Some slight revision in this section would achieve better flow.

Some specific minor points in the Introduction:

Line 83: clarify that these genera are in different taxonomic families. The wording now makes it seem like they are genera within a family.

Line 87: Again, give the family of the Sobemoviruses.

Line 93: Typo, "reaches"?

Lines 189-194: Some or all of this seems like it should be in Methods.

Line 324: the mention of rabies virus seems very out of context. If comparisons are useful to make to this virus that should be done in the Discussion.

Please use past tense.

Discussion - It would be really helpful to illustrate the seven periods that you have identified into a figure of sorts, such as a time line or similar.

Reviewer #2: I think the presentation of the paper may be improved, and the suggestions below could be taken into account:

1. Check the reference list, as the format is not uniform: Doi is given for some references and page number for other.

2. Line 141. Unclear. What means rice cultivation is localized?

3. Line 163 and on. Since the Material and Methods comes after the results, it would be good if right at the begging of the results the data sets on which the analyses are based were explained, not just referred as WA335 etc. Also, defined BETS.

4. Lines 184-185. I don’t find this information in Table 1.

5. The Discussion is very long and sometimes repetitive, what diffuses somewhat the message.

6. Also, the wealth of historical data provided contributes to explain, and complements, the hard-core results based on sequence analyses, but still is circumstantial evidence (cannot be otherwise) and should be so presented.

7. Line 358. Eragrostideae (I think this is the correct spelling) is a tribe within the Poaceae not a family. Thus I do not know if the sentence refers to hosts in other tribes of Poaceae or in other families related to Poaceae.

8. The Introduction does not indicate that finger millet (whatever it is, non -English-speaking readers would benefit from the botanical name) is a host of RYMV. A reference should be given.

9. Lines 362-363. Change to “ Strains specific to poaceae species….”

10. Line 381. Dispersal by boat, or rice introduced by boat?

11. Genomic regions used in analyses are sometimes identified by their product (e.g., coat protein sequences) and sometimes by the ORF (e.g., ORF4). As not everybody is familiar with the genome organization of this virus, it would help if some explanation was given on the non-CP regions analysed.

Reviewer #3: Author summary: "It also highlights the role of human transmission of pathogens, even vector-borne, and sheds light on the risk of transmission of RYMV and of other plant viruses from Africa to other continents." -- I wonder why the authors qualified this statement with "even vector-borne"? To me it seems like humans have played a massive role in the spread of many vector-borne viruses.

Introduction: The first paragraph seems a bit out of place. I was left wondering why I was being provided with all this information about transmission through direct contact when RYMV is (mostly?) vector-borne.

Lines 105-106: "Altogether, contact and soil transmission accounts for on-site survival of the virus and its local spread." -- Could you please clarify how this relates to vector-borne transmission? Can the virus persist without vector transmission?

Lines 130-131: "the phylogeographic reconstructions are not affected by the time dependent rate phenomenon" -- This does not seem like a defensible statement. Surely movement rates between different locations must to some extent fluctuate through time.

Line 163: "BETS" - please define BETS. Also in the same line its not clear what "increase in sample size" the authors are referring to.

Finally, for the comparative dispersal analysis, Trovao et al. (2015) previously performed an extremely elegant analysis examining why RYMV's diffusion rates vary through continental Africa and linked it to the intensity of rice production. I find it somewhat odd that these results where not discussed here. Disclosure: I have no affiliation with the Trovao et al. paper. I just think it deserves revisiting here.

PLOS authors have the option to publish the peer review history of their article (what does this mean? ). If published, this will include your full peer review and any attached files.

**Do you want your identity to be public for this peer review?** For information about this choice, including consent withdrawal, please see our Privacy Policy .

Reviewer #1: No

Reviewer #2: No

Reviewer #3: **Yes: ** David A. Rasmussen

---

## [Decision Letter · Decision Letter 1]

7 Apr 2025

PPATHOGENS-D-24-01977R1

Grains, trade and war in the multimodal transmission of Rice yellow mottle virus: an historical and phylogeographical retrospective

PLOS Pathogens

Dear Dr. Hebrard,

Thank you very much for your thorough and excellent responses to the reviewer comments. The reviewers and myself are ready to accept the manuscript. However, there are two small editing points it would be helpful for you to address first (detailed below). I appreciate that you all wish the work to be as high quality as possible, and these small edits will contribute to that quality. The suggested edits should require only a little time, perhaps just minutes. Once the ms. is returned, I will review it directly for full acceptance; it will not be sent out for further external review. 

Please submit your revised manuscript within 30 days Jun 06 2025 11:59PM, but preferably much sooner. The changes requested are small. Please include the following items when submitting your revised manuscript:

* A note that explains what changes you made.  You should upload this note as a separate file labeled 'Response to Reviewers'. **The note can be very short.** This file does not need to include responses to any formatting updates and technical items listed in the 'Journal Requirements' section below.

Please reach out if you have any questions. We look forward to receiving your updated, final manuscript.

Kind regards,

Carolyn M. Malmstrom

Guest Editor

PLOS Pathogens

Ronald Swanstrom

Section Editor

PLOS Pathogens

Sumita Bhaduri-McIntosh

Editor-in-Chief

PLOS Pathogens

orcid.org/0000-0003-2946-9497

Michael Malim

Editor-in-Chief

PLOS Pathogens

orcid.org/0000-0002-7699-2064

**Additional Editor Comments :**

Thank you for the extensive and thoughtful responses to the reviewer points. The manuscript is stronger and more accessible now, and well highlights the strength of the study, which will be of interest to many.

The reviewers and myself are ready to accept the ms. However, reviewer 3 has made a few textual suggestions of which it would be helpful if you could respond to two small but valuable points:

--"BETS analysis" still appears in the Results before it is defined/introduced.

--Lines 240-241: It might be helpful to note that the historical scenario described below is the most probable given the phylogeographic reconstruction - which would still acknowledge that there is some uncertainty in this reconstruction.

The two additional points made by reviewer 3 about whether material should be in Discussion or elsewhere are helpful points but I think the ms. can stand as is on those, so you need not respond to these, although it is fine if you do so.

The two points it would be helpful to address can be responded to in a few minutes and the ms. will be ready for full acceptance. I will handle the returned ms. directly and it will not be sent out for further external review.

Please let me know if you have any questions.

Best wishes, Carolyn

**Journal Requirements:**

1)  Thank you for stating "Nucleotidic sequences and associated metadata used in this study are available in the NCBI database." Please note that your Data Availability Statement is currently missing the DOI/accession number of each dataset OR a direct link to access each dataset. 

2) Please ensure that the funders and grant numbers match between the Financial Disclosure field and the Funding Information tab in your submission form. Note that the funders must be provided in the same order in both places as well. Currently, " National Commission for Science, Technology and Innovation (NACOSTI) " is missing from the Funding Information tab.

**Reviewers' Comments:**

Reviewer's Responses to Questions

**Part I - Summary**

Reviewer #1: The authors have addressed most of what I had recommended in my previous review.

Reviewer #2: I am happy with the revised version of this manuscript. All my comments to the original version have been addressed satisfactorily.

Reviewer #3: The authors have largely addressed my concerns about phylogeographic uncertainty and the molecular clock analysis. I just have a few additional comments about the text.

**Part II – Major Issues: Key Experiments Required for Acceptance**

Reviewer #1: None

Reviewer #2: No major issue

Reviewer #3: None

**Part III – Minor Issues: Editorial and Data Presentation Modifications**

Reviewer #1: None

Reviewer #2: No minor issue

Reviewer #3: I'm still a bit confused by the first paragraph of the Introduction. It makes many statements about other viruses before introducing RYMV. Maybe it would help if RYMV was first introduced and then explain how its biology contrasts with other viruses?

"BETS analysis" still appears in the Results before it is defined/introduced.

Lines 181-189. Multiple mechanisms are proposed for the lack of temporal signal in RYMV dataset but maybe this would fit better in the Discussion?

Lines 221-239: Likewise, maybe the discussion of sampling issues in phylogeography would fit better in the Discussion next to a discussion of other potential limitations?

Lines 240-241: It might be helpful to note that the historical scenario described below is the most probable given the phylogeographic reconstruction - which would still acknowledge that there is some uncertainty in this reconstruction.

PLOS authors have the option to publish the peer review history of their article (what does this mean? ). If published, this will include your full peer review and any attached files.

**Do you want your identity to be public for this peer review?** For information about this choice, including consent withdrawal, please see our Privacy Policy .

Reviewer #1: No

Reviewer #2: No

Reviewer #3: **Yes: ** David Rasmussen

**Figure resubmission:**
---

## [Editor Report · Decision Letter 2]

28 Apr 2025

Dear Dr Hebrard,

We are pleased to inform you that your manuscript 'Grains, trade and war in the multimodal transmission of Rice yellow mottle virus: an historical and phylogeographical retrospective' has been provisionally accepted for publication in PLOS Pathogens.

Best regards,

Carolyn M. Malmstrom

Guest Editor

PLOS Pathogens

Ronald Swanstrom

Section Editor

PLOS Pathogens

Sumita Bhaduri-McIntosh

Editor-in-Chief

PLOS Pathogens

orcid.org/0000-0003-2946-9497

Michael Malim

Editor-in-Chief

PLOS Pathogens

orcid.org/0000-0002-7699-2064

Thank you for addressing these last points. The responses are satisfactory, and I am pleased to be able to recommend full acceptance of the manuscript for publication.
---

## [Editor Report · Acceptance letter]

Dear Dr Hébrard,

We are delighted to inform you that your manuscript, "Grains, trade and war in the multimodal transmission of Rice yellow mottle virus: an historical and phylogeographical retrospective," has been formally accepted for publication in PLOS Pathogens.

Best regards,

Sumita Bhaduri-McIntosh

Editor-in-Chief

PLOS Pathogens

orcid.org/0000-0003-2946-9497

Michael Malim

Editor-in-Chief

PLOS Pathogens

orcid.org/0000-0002-7699-2064